# Impact of the Intake of Snacks and Lifestyle Behaviors on Obesity among University Students Living in Jeddah, Saudi Arabia

**DOI:** 10.3390/healthcare10020400

**Published:** 2022-02-21

**Authors:** Najlaa M. Aljefree, Israa M. Shatwan, Noha M. Almoraie

**Affiliations:** Food and Nutrition Department, Faculty of Human Sciences and Design, King Abdulaziz University, Building 43, Room 237, Level 2, Jeddah 3270, Saudi Arabia; eshatwan@kau.edu.sa (I.M.S.); nalmorie@kau.edu.sa (N.M.A.)

**Keywords:** snacks, obesity, lifestyle behaviors, young adults

## Abstract

Unhealthy eating habits increase the risk of obesity. This study investigated the association between obesity and the intake of snacks and lifestyle behaviors among university students in Saudi Arabia. The study included 662 students aged 18–29 years, studying at King Abdulaziz University. An online survey collected data on sociodemographic characteristics, height, and weight, to calculate body mass index (BMI), lifestyle behaviors, dietary habits, and snack intake. The prevalence of overweight and obese students was 18.6% and 12.7%, respectively. Sociodemographic characteristics and lifestyle behaviors had insignificant effects on obesity. Obese students consumed two meals daily and more cereals during breakfast. Non-obese students consumed more beverages at breakfast and had their daily meals with their families. The frequency of intake of snacks had an insignificant effect on obesity. However, obese students had a significantly higher intake of potato chips, popcorn, and biscuits, while non-obese students had a significantly higher intake of salads compared with obese students. Students consumed significantly less fruit and vegetables, chocolate, biscuits, nuts, and dairy products as snacks when inside the university compared to outside. To reduce obesity among students, universities should ensure access to healthy snacks, and provide health education programs to encourage healthy eating habits and lifestyles.

## 1. Introduction

The worldwide prevalence of obesity has greatly increased over the last few decades [1]. The prevalence rates of overweight and obese adults are estimated to be 25–50% and 13–50%, respectively, in the Arabian Gulf States [2]. Furthermore, obesity is prevalent in Saudi Arabia. The national rates of overweight and obesity (body mass index (BMI) ≥ 25 kg/m^2^) in men and women aged 18–29 years were 30.4% and 10.2%, respectively [3]. A previous survey found that among Saudi adolescents aged 13–18 years old, 26.6% and 10.6% were overweight and obese, respectively [4]. Obesity is a severe health concern; a scoping review of risk behavior interventions in young men aged 20–39 years showed an obesity incidence rate of 29% [5]. Moreover, over the last several decades, there has been an increase in the incidence of obesity among Saudi youth [6,7,8]. Obesity is a complex condition that is affected by various controllable and uncontrollable factors [9]. Obesity is linked to several controllable variables, including physical inactivity, sedentary lifestyles, and poor dietary habits [9]. Unhealthy dietary patterns and physical inactivity are two of the most common factors 4that might have a negative impact on young adults’ weight status and, thereby, on adult health [10,11]. University is a critical age for young adults in terms of dietary choices and weight gain [12,13,14]. Several studies have presented differences between those who attend university and those who do not, where those who attend university gained more weight compared with those who did not [14,15].

‘Snacking’ is a term used to describe the intake of food and beverages, such as chips, chocolates, and soft drinks. Snacking rates among young individuals, particularly university students, have been studied globally [16]. Snacking is frequently linked to negative health outcomes and poor dietary habits, and has thus been considered a contributing factor in overweight or obese individuals [17]. The effects of snacking are difficult to study owing to the diverse strategies used to study snacking and the various definitions of snacking that have been employed in prior research [18]. Furthermore, studies that have assessed the association between snacking and BMI have generated contradictory results. The common perception of snack foods is that they are rich in fat and sugar, making them unhealthy and inappropriate for healthy eating habits [19]. The terms ‘snacking’ and ‘snack’ are extensively used; however, they have not been given a general definition. Snacking, in general, may be described as the consumption of food or drinks between meals [20] and food consumption based on time requirements [21]. According to Drummond et al. [22], snacking may not always predispose an individual to obesity, as a higher eating frequency may be beneficial in terms of body weight regulation and energy balance; snacking could decrease calorie intake when replacing main meals. Some authors have claimed that there is an inverse relationship between the prevalence of obesity and high meal frequency [23,24,25]. Moreover, the timing of snacking may have an impact on metabolism and energy expenditure. This is because evening eating reduces diet-induced thermogenesis relative to morning and afternoon snacking, and overnight snacking reduces whole-body fat oxidation [26,27,28]. Regardless of the possible link between snack intake and obesity, the frequency of snacking among young adults has not been thoroughly examined. Therefore, this study aimed to investigate the association between obesity and the intake of snacks and lifestyle behaviors among university students in Saudi Arabia. Furthermore, the frequencies of consumption of some common snack items inside and outside of the university were also assessed.

## 2. Methods

### 2.1. Study Participants

An online survey was administered to students at the King Abdulaziz University via a web link shared over the official e-mail system of the university in Jeddah, Saudi Arabia, between January and March 2021. The study used a sample of full-time university students from all academic years. A total of 662 university students aged 18–29 years across both sexes made up the study sample. According to the Deanship of Admission and Registration at the King Abdulaziz University, the total population of students in 2019 was 80,000; given this number, the sample size calculator indicated that the sample size needed to achieve sufficient statistical power was 659, which was based on a 5% margin of error, 99% confidence level and response rate of 50% [29].

### 2.2. Questionnaire Design

This cross-sectional study was approved by the Biomedical Ethics Research Committee of the King Abdulaziz University and was conducted in accordance with the Declaration of Helsinki (reference no. 9–21). The invitation to participate in the survey was forwarded to students via university e-mail. The e-mail and the introduction to the online questionnaire explained the aim of the study and provided guarantees of anonymity throughout the investigation. The questionnaire was written in Arabic. The questions included in the survey were developed using data from previous studies that examined the intake of snacks and lifestyle behaviors of university students in different countries [30,31,32]. Before the online submission, the questionnaire was piloted to verify whether the items were clearly understood by the participants. Subsequently, the final questionnaire was developed and distributed using Google Forms.

### 2.3. Demographic Data and Anthropometric Measurements

Each participant was asked to complete a questionnaire on his or her sociodemographic characteristics. It included general information on age in years (18–21, 22–25, and 26–29), sex (male and female), social status (married, single, divorced, and widowed), nationality (Saudi and non-Saudi), academic year (first, second, third, fourth, fifth, and sixth), and monthly household income (<5000, 5000–15,000, >15,000 Saudi Riyals (SR)). Self-reported height and weight were used in this study to calculate BMI, which was calculated as body weight in kilograms divided by height in meters squared (kg/m^2^). Participants were classified into four groups: underweight (BMI < 18.5 kg/m^2^), normal weight (BMI 18.5–24.9 kg/m^2^), overweight (BMI 25–29.9 kg/m^2^), and obese (BMI ≥ 30 kg/m^2^).

### 2.4. Lifestyle Behaviors

A questionnaire was developed to identify lifestyle behaviors. A total of six items were queried: (1) how many times per week they engage in physical activity or sports (1–2 times a week, 3–4 times a week, >4 times a week, and none), (2) the usual time spent watching TV or using a mobile phone per day (2–4 h, 5–7 h, and >7 h), (3) if they prefer to watch food advertisements (yes or no), (4) how long they usually sleep (<5 h, 5–8 h, and >8 h), (5) when they usually sleep (at night and in the daytime), and (6) whether they suffer from sleep disorders (yes or no).

### 2.5. Dietary Habits and the Intake of Snacks Data

This section discusses nine survey questions about participants’ dietary habits. These questions queried the following: (1) the number of main meals consumed per day (1, 2, or 3), (2) the number of times breakfast is eaten per week (rarely, 1–2 times, 3–4 times, daily), (3) whether breakfast is eaten at the university (yes, no, or sometimes), and (4) type of breakfast meals (sandwiches or savory pastries, sweets or biscuits, dairy products, fruits and vegetables, beverages, and cereal), (5) whether meals were eaten alone or with family (always alone, alone 1–2 times per week, alone 3–4 times per week, and daily with family). Questions about snacks, included (6) frequency of snack intake per week (daily, 3–4 times per week, 1–2 times per week, rarely, and never), and (7) time of snacks (not specified, mid-morning, between breakfast and lunch, between lunch and dinner, and after dinner). Furthermore, they were asked about (8) activities while eating snacks (television viewing, studying, playing video games, and no specific activity) and (9) the type of snacks they eat inside and outside the university (fruits and vegetables, chocolate, nuts, potato chips, popcorn, doughnuts, dairy products, sandwiches, biscuits, and salad).

### 2.6. Statistical Analysis

Data were analyzed using SPSS version 28 (IBM Corp., Armonk, NY, USA). Descriptive analysis was conducted, including frequency and percentage. Comparisons between categorical data were performed using the Chi-square test. Logistic regression analyses were used to study the differences in consumption of some snack items between settings inside and outside the university. Underweight and normal weight participants were combined, and overweight and obese were also combined in order to perform logistic regression. All differences were considered significant with *p* values < 0.05.

## 3. Results

Table 1 presents the sociodemographic characteristics of the study participants according to their BMI status. Data were obtained from a total sample of 662 university students, 371 of which were 18–21 years old (56%), 272 of which were 22–25 years old (41.1%), and 19 of which were 26–29 years old (2.9%); and comprised 103 men (15.6%) and 559 women (84.4%). Almost all of the participants were single (92.1%) and Saudis (93.2%). There were insignificant associations between sociodemographic characteristics and obesity. In addition, the prevalence of underweight, normal weight, overweight, and obesity was reported as 20.4%, 48.3%, 18.6%, and 12.7% among study participants, respectively.

Table 2 shows the lifestyle behaviors of the study population according to BMI status. There were no significant differences between non-obese students and obese students in terms of physical activity, hours spent on watching TV or using a mobile phone, watching food advertisements, and sleeping habits.

Table 3 presents the meals and breakfast patterns of the study populations according to BMI status. There was a significant difference in the number of meals per day consumed between the observed groups. Although the majority of both non-obese and obese students consumed two meals daily, the percentage of obese students was higher than that of non-obese students (*p* = 0.004). The results for items consumed during breakfast showed significant differences in cereal (*p* = 0.008) and beverage consumption (*p* = 0.04). Obese students had higher consumption of cereals than non-obese students did, whereas non-obese students had higher intake from beverages. All students were more likely to eat a meal with their families daily; however, the percentage was significantly higher among non-obese students (*p* = 0.03).

Table 4 displays the intake of snacks in the study population according to BMI status. Among university students, 35% consumed snacks on a daily basis, whereas almost 25% consumed snacks 3–4 times per week, although the effect of frequent snack intake on obesity was insignificant. Several food items have been reported among the most frequent snacks consumed by university students, including chocolate (70%), potato chips (50%), biscuits (20%), fruits and vegetables (40%), nuts (55%), popcorn (40%), and salad (25%); however, some of these food items showed insignificant effects on obesity. Obese students consumed potato chips (56.8% vs. 45.2%; *p* = 0.004), popcorn (41.7% vs. 33.3%; *p* = 0.04), and biscuits (20.9% vs. 14.3%; *p* = 0.04) more frequently than non-obese students, whereas salad was consumed more frequently among non-obese students than obese students (25.2% vs. 18%; *p* = 0.04). 

Table 5 displays the differences in frequency of consumption of some snack items outside and inside the university. Significant differences were detected in the frequencies of consumption of fruit and vegetables, chocolate, biscuits, nuts, and dairy products outside and inside the university (*p* < 0.001). Inside university, students consumed less fruit and vegetables (2.7% vs. 40.6%), chocolate (53.5% vs. 70.7%), biscuits (3.5% vs. 16.3%), nuts (2% vs. 55%), and dairy products (1.8% vs. 12.7%), compared to outside university. The odds of consumption of fruit and vegetables among students inside university was 0.04 (95% CI 0.02–0.06), chocolate 0.47 (95% CI 0.38–0.59), biscuits 0.18 (95% CI 0.11–0.29), nuts 0.01 (95% CI 0.009–0.02), and dairy products 0.12 (95% CI 0.06–0.23).

## 4. Discussion

Obesity is a well-known global health issue. The prevalence of obesity has increased significantly across all age groups in recent decades. Several factors contribute to an increased risk of obesity, including unhealthy eating habits, such as the intake of energy-dense snacks that are rich in sugar and fat. The consumption of snacks has been linked to an increased risk of obesity. This study investigated the association between obesity and the intake of snacks and lifestyle behaviors among university students in Saudi Arabia. Results showed a high prevalence of overweight (18.6%) and obesity (12.7%) among the study participants. In addition, there were no significant differences in lifestyle behaviors, including physical activities, watching TV or using a mobile phone, watching food advertisements, sleeping hours, and time of sleep, between obese and non-obese students. Furthermore, the results showed that obese students consumed two meals daily and had higher consumption of cereals during breakfast, whereas the non-obese students had higher intake from beverages at breakfast, and had their daily meals with their families. Moreover, obese students consumed popcorn, biscuits, and potato chips as snacks more frequently than non-obese students did, whereas non-obese students had higher consumption of salads as snacks compared to obese students. In addition, students consumed significantly less fruit and vegetables, chocolate, biscuits, nuts, and dairy products as snacks inside the university compared with outside the university.

In recent decades, there has been a rising trend of overweight and obesity worldwide, especially among young adults. Previous studies have shown that university students gain extra weight in their early years at university [30]. Evidence from developing countries indicates a high burden of overweight and obesity among the younger generations [33]. For example, the prevalence of overweight and obesity among university students was relatively high in Malaysia (15.9% and 5.2%) [34], India (26.8% and 10.7%) [35], Egypt (36.9% and 12.5%) [36], and Oman (26.7% and 1.5%) [37], respectively. In Saudi Arabia, along with other Gulf countries, significant economic expansion has negatively influenced peoples’ health. It has affected dietary habits, as the consumption of foods rich in fat and sugar, and fast food, has increased with a significant decrease in physical activity, thus increasing the burden of obesity and related diseases. The prevalence of obesity among adults in the Gulf countries ranges from 22% to 34.1% among men and from 26.1% to 44% among women [38]. Among university students in Saudi Arabia, a study of female students showed that 31.4% were overweight and 16.5% were obese [39]. Similarly, a study conducted in Riyadh showed that 31% of university students were overweight and 23.3% were obese [40], which is higher than the rates reported in this study.

Physical inactivity is considered a leading cause of obesity worldwide. Physical activity has declined globally during early adulthood in university students [41]. In this study, no significant association was found between obesity and physical activity; however, more than one-third of university students did not engage in any physical activities. Several studies have reported a high prevalence of inactivity among university students. A study in Venezuela of 314 college students showed that 18% of men and 35% of women found that they were not sufficiently active [42]. Similarly, 65% of Saudi female university students reported not practicing any physical activities during the week [39]. In addition, a study among university students from 22 different countries showed that 22.7% of students reported practicing low levels of physical activity [33]. The study also reported a significant association between physical inactivity and obesity [33]. Other studies reported similar results [43,44]; however, this was not the case in this study. We should note that practicing physical activity was not affected by the COVID-19 pandemic during the time of data collection for this study as there were no restrictions on gyms and health clubs that prevented people from engaging in physical activities.

Despite the lack of physical activity, this study showed that almost half of the students spent approximately 7 h per day watching TV or using their mobile phones, which worsens the issue of weight gain and inactivity. Another study in Saudi Arabia reported that 42% of university students spent 4–6 h per day watching TV or using computers. The study also reported no significant relationship between these factors and obesity status [45]. Sleeping patterns and their relationship to weight gain have recently become a focus of research. In this study, almost a quarter of university students slept for long durations (>8 h), with one-third of them sleeping during the daytime. In addition, the results showed no significant association between sleeping patterns and obesity. A previous study indicated that longer sleep duration (>9 h) was significantly associated with an increased risk of obesity in adults, while shorter sleep duration had no effect on obesity [46]. In contrast, other studies showed no evidence that sleep duration affects weight gain [47,48], which is consistent with the findings of this study.

Among university students worldwide, several unhealthy eating habits have been identified, including skipping breakfast, frequent consumption of snacks, and high intake of fast food [49,50]. In a previous study, the intake of breakfast was associated with a decreased risk of obesity [51,52]. In this study, the intake of breakfast in general or inside university was not associated with obesity; however, almost one-quarter of university students stated that they rarely eat breakfast. These findings correspond with the findings of a study from United States, where approximately 25% of young adults skipped breakfast [53]. Previous evidence has shown that skipping breakfast is linked to increased weight gain [53,54]. Regarding the type of breakfast, this study showed that obese students had higher consumption of cereals, while non-obese students had a higher intake of beverages. A previous national study in the United States reported that the consumption of ready-to-eat cereals was associated with a low risk of obesity [53]. That study explained this finding as follows: the consumption of cereals is usually associated with a high intake of dairy products, and cereals are usually fortified with vitamins, minerals, and fibers [53]. However, in our study, the consumption of cereals was associated with weight gain, which might be explained by the increase in added sugar in the types of cereals that university students consume. Moreover, the results of this study showed a significant difference in the number of meals per day consumed by students. Although both groups consumed two meals daily, the percentage of obese students doing this was significantly higher than that of non-obese students. Similar studies from Saudi Arabia and Lebanon showed that the majority of students (55.7% and 47.9%, respectively) consumed two meals per day [50,55]. In contrast, a previous study reported that the consumption of one or two meals per day was associated with a lower risk of obesity than the consumption of three meals per day. The study suggested that reducing the number of main meals per day (to <3) would benefit in reducing hunger and feeling full [56]. Likewise, a study conducted in China showed that the vast majority of university students consume three meals per day, and the prevalence of obesity among them was only 2.9% [57]. Hence, we can conclude that the findings are mixed regarding the frequency of meals per day and its relation to weight gain. Furthermore, this study showed that all students had daily meals with families; however, the percentage was significantly higher among non-obese students. In contrast, another study in Saudi Arabia reported that >60% of university students had their meals with their families on a daily basis; however, a large number of those students were considered obese [45]. In fact, eating meals with family, especially for the younger generation, might have several benefits, including having a wide variety of healthy food to choose from, and positive role-modeling from their parents’ example and influence in consuming healthy food [58]. This may explain our finding that non-obese students who consumed their daily meals with their families were more inclined to choose healthier food options than their obese counterparts. Previous studies have shown that students who consume their meals away from home and families tend to be obese as they usually consume food high in total calories, fat, and sugar, such as fast-food meals [59].

This study showed that one-third of university students stated that they eat snacks daily, while about a quarter of students said they eat snacks to 3–4 times per week. Less than 15% stated that they rarely eat snacks. Another study in Saudi Arabia reported a similar intake of snacks, as almost 30% of university students were eating snacks on a daily basis [50]. In comparable studies, the intake of daily snacks was prevalent among 50% of Lebanese university students [55], and 30% of Chinese female students; however, it was lower among Chinese male students (11%) [57]. In Malaysia, a study reported that almost half of the students were eating snacks daily [49]. Likewise, 54% of Indian university students eat snacks daily [60]. In addition, this study showed no significant correlation between eating snacks and obesity. Previous studies were aligned with our results [45,49,61]. In contrast, other studies have reported a positive association between snack intake and weight gain [62,63]. Surprisingly, some studies have shown that frequent snacking might lead to weight loss [50]. Thus, we can conclude that previous research examining the correlation between eating snacks and obesity has generated contradictory results. It is difficult to examine the effect of eating snacks on obesity, as there are numerous strategies used to study snacking and the different definitions of snacking that have been employed in prior research [18]. Moreover, almost half of the students in our study stated that they were eating snacks while watching TV or using mobile phones; however, there was no significant difference between the observed groups. Equivalent studies showed similar results, as 60% of Malaysian students and 51% of Indian students were eating snacks while watching TV [49,60]. In addition, this study indicated that almost 80% of students had no specific time to eat snacks, which is nearly identical to the finding of the Indian study, in which 78% of students had no specific time to eat snacks [60].

Regarding the type of snacks consumed by university students in this study, a number of food items that are high in fat and sugar have been reported among the most frequent snacks consumed by university students, including chocolate (70%), potato chips (50%), and biscuits (20%). Contrarily, a number of healthy food items have also been reported among the most frequent snacks consumed by university students, including fruits and vegetables (40%), nuts (55%), popcorn (40%), and salad (25%). In addition, obese students had a significantly higher intake of potato chips, popcorn, and biscuits than non-obese students did, whereas non-obese students had a significantly higher intake of salads than obese students did. The high intake of potato chips among obese students has also been reported in previous research [64]. A recent study conducted in the USA on university students showed that the types of food items highly consumed as snacks were potato chips and chocolates by 72% and 64% of students, respectively [64]. This finding is consistent with our findings. Previous studies have shown that the types of snacks consumed by young adults are usually characterized by high amounts of fat, sugar, and total calories, and are considered unhealthy eating habits that might contribute to weight gain [18,19]. Prior evidence suggests that chocolates/candies, cookies, biscuits, donuts, potato chips, and nuts are among the most common types of snacks consumed by university students [49,61,62,65]. Remarkably, our results also showed that almost 40% of university students consumed fruits and vegetables as snacks. Indeed, evidence from Saudi Arabia has demonstrated low consumption of fruits and vegetables [3,66]. Diets rich in fruits and vegetables are known to be associated with weight reduction owing to the high concentration of fiber and water, as well as low calories and fat in raw fruits and vegetables, which is consistent with our findings that non-obese students consumed more salads than obese students. Surprisingly, the intake of popcorn was significantly higher among obese students, even though it is rich in fibers, antioxidants such as phenolic acids, and low in total calories [67], which explains why it is considered a healthy snack. One possible reason for the link between the intake of popcorn and obesity could be the added flavors such as chocolate or caramel, which might increase the total calories and sugar in the popcorn, thus increasing weight gain.

Moreover, the findings of this study indicated that eating snacks within a university setting might affect the types of snacks students choose, and whether they consider eating healthy or unhealthy snacks. The results of this study indicated that the intake of fruits and vegetables (2.7% vs. 40.6%), nuts (2% vs. 55%), and dairy products (1.8% vs. 12.7%) were significantly lower inside universities than outside universities, respectively. Furthermore, the intake of chocolate was also significantly lower inside university compared to outside university (53.5% vs. 70.7%, respectively); however, there is still high consumption irrespective of the location. Students usually spend a considerable amount of time at university; hence, the food environment on campus, including the availability of healthy eating choices such as healthy meals and snacks with reasonable prices, has an important role in developing healthy eating behaviors for young adults [68]. Prior studies have shown that the available food and snacks in universities are characterized by food items high in calories, fat, and sugar [69,70]. A study conducted in Australia illustrated that chocolates, potato chips, and sugar-sweetened beverages were among the most commonly available food items at university campuses [70], which affects weight gain and dietary behaviors among university students. This might explain our finding that the intake of healthy snacks, including fruits and vegetables, nuts, and dairy products, was significantly lower inside the university than outside the university.

This study had several limitations. The cross-sectional study design could not determine causality. Self-reported weight and height numbers are considered a crucial limitation for this study; however, this method is frequently used in previous studies, especially for data collected through an online survey, because of the COVID-19 pandemic. Furthermore, selection bias might be an issue, as data were collected through an online survey owing to restrictions during the COVID-19 pandemic. In addition, the results are not generalizable to all Saudi university students because data were collected from only one university. In addition, recall bias might be of concern, as some responses required remembering. Some factors that might affect the intake of snacks, including emotional eating and level of stress, were not measured in this study. Contrarily, the study focused on factors affecting the burden of obesity among young adults, which is a well-known public health issue in Saudi Arabia. The study also compared the intake of some common snack items among students inside and outside university to reflect the food environment at university campus, which is a strong point of this study.

## 5. Conclusions

In the current study, we investigated lifestyle behaviors and types of snacks related to risk of obesity among university students. Snacks consumed contribute to the intake of daily energy and nutrients, and thus affect the overall quality of the diet. The most highly consumed snacks among the study participants were considered unhealthy items except for fruits, vegetables, and nuts. Consumption of potato chips, popcorn, and biscuits as snack items were highly associated with obesity risk, whereas salad intake was associated with a decreased obesity risk. The university environment was associated with reduced consumption of healthy types of snacks, such as fruits, vegetables, nuts, and dairy products. Health education plans and programs among university students are crucial to encourage healthy eating habits, including eating breakfast and frequent consumption of healthy snacks. Universities should also focus on the food environment of campuses and ensure access to healthy snacks at reasonable prices, which is essential in developing healthy dietary behaviors, decreasing the high rates of obesity in young adults, and ultimately improving public health.

## Figures and Tables

**Table 1 healthcare-10-00400-t001:** Sociodemographic characteristics of the study participants according to body mass index status.

Characteristics	Total (n = 662)	Non-Obese Students (n = 456)	Obese Students (n = 206)	*p* Value
Gender				
Male	103 (15.6)	63 (13.8)	40 (19.4)	0.08
Female	559 (84.4)	393 (86.2)	166 (80.6)	
Age				
18–21	371 (56)	267 (58.6)	104 (50.5)	0.11
22–25	272 (41.1)	175 (38.4)	97 (47.1)	
26–29	19 (2.9)	14 (3.1)	5 (2.4)	
Marital status				
Single	610 (92.1)	422 (92.5)	188 (91.3)	0.18
Married	47 (7.1)	29 (6.4)	18 (8.7)	
Divorced	5 (0.8)	5 (1.1)		
Nationality				
Saudi	617 (93.2)	420 (92.1)	197 (95.6)	0.13
Non-Saudi	45 (6.8)	36 (7.9)	9 (4.4)	
Academic years				
First	87 (13.1)	64 (14)	23 (11.2)	0.84
Second	125 (18.9)	84 (18.4)	41 (19.9)	
Third	139 (21.0)	97 (21.3)	42 (20.4)	
Fourth	201 (30.4)	139 (30.5)	62 (30.1)	
Fifth	56 (8.5)	38 (8.3)	18 (8.7)	
Sixth	54 (8.2)	34 (7.5)	20 (9.7)	
Monthly household income (SR):				
<5000	135 (20.4)	93 (20.4)	42 (20.4)	0.21
5000–15,000	298 (45)	196 (43)	102 (49.5)	
>15,000	229 (34.6)	167 (36.6)	62 (30.1)	

The Chi-square test was used to examine difference between two groups.

**Table 2 healthcare-10-00400-t002:** Lifestyle behaviors of the study populations according to body mass index status.

Lifestyle Behaviors	Non-Obese Students (n = 456)	Obese Students (n = 206)	*p* Value
Physical activities			
None	194 (42.5)	78 (37.9)	0.56
1–2 times a week	133 (29.2)	62 (30.1)	
3–4 times a week	75 (16.4)	42 (20.4)	
>4 times a week	54 (11.8)	24 (11.7)	
Watching TV or using a mobile phone			
2–4 h	76 (16.7)	40 (19.4)	0.66
5–7 h	202 (44.3)	86 (41.7)	
<7 h	178 (39)	80 (38.8)	
Watching food advertisements			
Yes	151 (33.1)	83 (40.3)	0.07
No	305 (66.9)	123 (59.7)	
Sleeping hours			
<5 h	22 (4.8)	9 (4.4)	0.73
5–8 h	307 (67.3)	145 (70.4)	
>8 h	127 (27.9)	52 (25.2)	
Time of sleep			
At night	303 (66.4)	151 (73.3)	0.08
In the daytime	153 (33.6)	55 (26.7)	
Sleeping disorders			
Yes	230 (50.4)	99 (48.1)	0.61
No	226 (49.6)	107 (51.9)	

The Chi-square test was used to examine difference between two groups.

**Table 3 healthcare-10-00400-t003:** Meals and breakfast pattern of the study populations according to body mass index status.

	Non-Obese Students (n = 456)	Obese Students (n = 206)	*p* Value
Number of meals			
1 Meal/day	65 (14.3)	14 (6.8)	0.004
2 Meals/day	258 (56.6)	141 (68.4)	
3 Meals/day	133 (29.2)	51 (24.8)	
Eating breakfast			
Rarely	95 (20.8)	50 (24.3)	0.67
1–2 times/week	76 (16.7)	30 (14.6)	
3–4 times/week	106 (23.2)	43 (20.9)	
Daily	179 (39.3)	83 (40.3)	
Eating breakfast at university			
Yes	102 (22.4)	33 (16)	0.09
No	169 (37.1)	74 (35.9)	
Sometimes	185 (40.6)	99 (48.1)	
Type of breakfast			
Cereal	75 (16.4)	53 (25.7)	0.008
Sandwiches or savory pastries	336 (73.3)	138 (67)	0.09
Sweets and biscuits	34 (7.5)	22 (10.7)	0.27
Dairy products	77 (16.9)	44 (21.4)	0.19
Fruits and vegetables	58 (12.7)	30 (14.6)	0.53
Beverages	112 (24.6)	36 (17.5)	0.04
Meals with family			
Always alone	51 (11.2)	18 (8.7)	0.03
1–2 times/week	78 (17.1)	49 (23.8)	
3–4 times/week	115 (25.2)	63 (30.6)	
Daily with family	212 (46.5)	76 (36.9)	

The Chi-square test was used to examine difference between two groups.

**Table 4 healthcare-10-00400-t004:** The intake of snacks of the study populations according to body mass index status.

	Non-Obese Students (n = 456)	Obese Students (n = 206)	*p* Value
Frequency of eating snacks			
Daily	153 (33.6)	71 (34.5)	0.13
1–2 times/week	89 (19.5)	57 (27.7)	
3–4 times/week	130 (28.5)	48 (23.3)	
Rarely	76 (16.7)	27 (13.1)	
Never	8 (1.8)	3 (1.5)	
Time of snacking			
Not specified	367 (80.5)	166 (80.6)	0.95
Mid-morning	31 (6.8)	16 (7.8)	
Between breakfast and lunch	13 (2.9)	5 (2.4)	
Between lunch and dinner	34 (7.5)	13 (6.3)	
After dinner	11 (2.4)	6 (2.9)	
Activity during snacking			
Watching TV or using a mobile phone	202 (44.3)	95 (46.1)	0.25
Studying	59 (12.9)	17 (8.3)	
Video games	25 (5.5)	16 (7.8)	
No specific activity	170 (37.3)	78 (37.9)	
Types of snacks			
Fruits and vegetables	186 (40.8)	83 (40.3)	0.93
Chocolate	317 (69.5)	151 (73.3)	0.35
Nuts	244 (53.5)	121 (58.7)	0.23
Potato Chips	151 (45.2)	117 (56.8)	0.004
Popcorn	152 (33.3)	86 (41.7)	0.04
Doughnuts	73 (16)	26 (12.6)	0.29
Dairy products	56 (12.3)	28 (13.6)	0.71
Sandwiches	7 (1.5)	2 (1)	0.72
Biscuits	65 (14.3)	43 (20.9)	0.04
Salad	115 (25.2)	37 (18)	0.04

Chi-square test used to examine difference between two groups.

**Table 5 healthcare-10-00400-t005:** Differences in consumption of snack items inside and outside the university.

Snack Items	Odds Ratio (95% Confidential Interval)	*p* Value
Fruit and vegetables Outside university vs. inside university	0.04 (0.02–0.06)	<0.001
Chocolate Outside university vs. inside university	0.47 (0.38–0.59)	<0.001
Biscuits Outside university vs. inside university	0.18 (0.11–0.29)	<0.001
Nuts Outside university vs. inside university	0.01 (0.009–0.02)	<0.001
Potato chips Outside university vs. inside university	0.87 (0.70–1.08)	0.21
Doughnuts Outside university vs. inside university	0.92 (0.68–1.26)	0.64
Dairy Outside university vs. inside university	0.12 (0.06–0.23)	<0.001
Sandwiches Outside university vs. inside university	1.68 (0.73–3.87)	0.22

Logistic regression was used to examine the effect of outside versus inside university on consumption snack items.

## Data Availability

The datasets generated and/or analyzed during this study are not publicly available owing to use of data for further publications, but are available from the corresponding author on reasonable request.

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
