# Peer review of "Impact of the Intake of Snacks and Lifestyle Behaviors on Obesity among University Students Living in Jeddah, Saudi Arabia"

_healthcare, 2022, doi:10.3390/healthcare10020400_

Round 1

Reviewer 1 Report

The paper presents an assessment of snack consumption and lifestyle behaviors among at the King Abdulaziz University students (Jeddah, Saudi Arabia) and their contribution to the increasing rates of obesity.

I have some reservations about the questions in the survey. In the methodology, the missing information is whether the questionnaire concerned only full-time students. Were also part-time (weekend) students participating in the survey? It should be clear.

L103-104. The authors should approach this question carefully and explain in the discussion what restrictions related to the pandemic were at that time. Were there, for example, closed gyms etc., which made it difficult to be physically active?

L111-112. ‘the number of times breakfast is eaten per week’. In survey is lack answer ‘everyday’ or ‘5 times and more’ Please explain.

L114 and Table 3. Answer ‘sandwiches or pastries’. ‘Pastry’ is a sweet, baked food. Why this kind of food not classified to group ‘sweets and biscuits’ or alone? If the authors separated these two types of food, it might turn out that obese students consume more pastries than sandwiches?! This is two different type of food.

L115-116. ‘always alone’ and ‘alone daily’ It isn’t the same? It is mistake? Please explain.

L145. h spent?

Table 5. Authors of the questionnaire (Methods section) did not ask students about eating snacks at university or outside (line 121-123). So what are the data in Table 5 based on?

L366-367. This sentence should be supported by a questionnaire. The questionnaire did not include eating of snacks inside and outside the university.

L374. sleeping h?

Author Response

Reviewer 1:

The paper presents an assessment of snack consumption and lifestyle behaviours among at the King Abdulaziz University students (Jeddah, Saudi Arabia) and their contribution to the increasing rates of obesity.

  • I have some reservations about the questions in the survey. In the methodology, the missing information is whether the questionnaire concerned only full-time students. Were also part-time (weekend) students participating in the survey? It should be clear.

We thank the Reviewer for this comment and the chance to clarify. In our study, we have only included full-time students, thus we have added “full-time university students” under study participants section (see lines 79, page 2).

  • L103-104. The authors should approach this question carefully and explain in the discussion what restrictions related to the pandemic were at that time. Were there, for example, closed gyms etc., which made it difficult to be physically active?

We thank the reviewer for the valuable comments. We collected our data at a late stage of the pandemic between January and March 2021 and there was no lockdown in Saudi Arabia at this period. In fact, the percentage of COVID vaccination was very high in Saudi Arabia as almost 85% of the population took 2 doses during the time of data collection. Hence, there was no impact of COVID pandemic on the physical activity levels among study subjects. we have added a sentence to the discussion, as follows: “We should note that practicing physical activity was not affected by the COVID-19 pandemic during the time of data collection for this study as there were no restrictions on gyms and health clubs that prevented people from engaging in physical activities.” (see lines 256-259, page 8).

  • L111-112. ‘the number of times breakfast is eaten per week’. In survey is lack answer ‘everyday’ or ‘5 times and more’ Please explain.

We thank the Reviewer for this comment. We have added the comma which was missing to indicate that the option for everyday (daily) is included. (see line 128, page 3).

  • L114 and Table 3. Answer ‘sandwiches or pastries. ‘Pastry’ is a sweet, baked food. Why this kind of food not classified to group ‘sweets and biscuits’ or alone? If the authors separated these two types of food, it might turn out that obese students consume more pastries than sandwiches?! This is two different type of food.

We thank the Reviewer for this comment and the chance to clarify. We meant by pastries savoury pastries such as croissant but not the sweet pastries. Hence, we change it to sandwiches and savoury pastries for more clarification (see line 130, page 3) and in Table 3 page 5.

  • L115-116. ‘always alone’ and ‘alone daily’ It isn’t the same? It is mistake? Please explain. daily Meals with family

We thank the Reviewer for this comment. We removed the word “alone” and add “daily with family” (see line 132, page 3). We also added “daily with family” to Table 3.

  • h spent?

We thank the Reviewer for this comment. h refers to hours spent on watching TV. It has been changed (see line 165, page 4).

  • Table 5. Authors of the questionnaire (Methods section) did not ask students about eating snacks at university or outside (line 121-123). So what are the data in Table 5 based on?

We asked in the questionnaire about the type of snacks they eat inside and outside university in separate questions but as they are both the same items and for clarification we have added “inside and outside the university” to the question as the following: “the type of snacks they eat inside and outside the university (fruits and vegetables, chocolate, nuts, potato chips, popcorn, doughnuts, dairy products, sandwiches, biscuits, and salad).” Therefore, the data in table 5, is the differences in consumption of snack items in and out university. (see line 137-138, page 3).

  • L366-367. This sentence should be supported by a questionnaire. The questionnaire did not include eating of snacks inside and outside the university.

We thank the Reviewer for this observation. We have addressed this question in the previous comment.

  • sleeping h?

We thank the Reviewer for this comment. h refers to sleeping hours. It has been deleted when we rewrote the conclusion section.

Reviewer 2 Report

Comments for Authors

Thank you for the opportunity to review the article.
There are a several issues that need to be improved before publication, details in the attachment.

Author Response

Reviewer 2:

Thank you for the opportunity to review the article.

There are a several issues that need to be improved before publication, details in the attachment.

  1. What is the purpose of the study?

- In abstract line 8-10: "This study investigated the association between obesity and the intake of snacks and lifestyle behaviours among university students in Saudi Arabia".

- At the end of the introduction, line 67-70: ... "this study aimed to assess the intake of snacks and lifestyle behaviours in university students and their contribution to the increasing rates of obesity. Furthermore, the frequencies of consumption of some common snack items inside and outside of the university were also assessed "

- Line 190-191: "This study investigated the intake of snacks and lifestyle behaviours among university students and their associations with obesity".

We thank the reviewer for the valuable comments. The aim of the study has been changed in the introduction, abstract, and discussion sections as follow “This study investigated the association between obesity and the intake of snacks and lifestyle behaviours among university students in Saudi Arabia.” (see lines 8-9, page 1 and lines 76-77, page 2, 224-225, page 7).

  1. Abstract, line 12-13: please remove the information about the software used.

We thank the Reviewer for this comment. Information about software has been removed.

  1. Keywords: phrases like university students; Saudi Arabia are not specific

We thank the reviewer for the valuable comments. university students and Saudi Arabia have been deleted from the key words and young adults has been added.

  1. Methodology:

- line 75-76, the authors state: "The study used a random sample of university students from all academic years". How was the sampling carried out, by algorithm, others?

We thank the reviewer for this comment and the chance to clarify. We removed word "random". We meant by "random" that we sent questionnaire through university e-mail to all university students. It has been changed to the following” An online survey was administered to students at the King Abdulaziz University via a web link shared over the official e-mail system of the university in Jeddah, Saudi Arabia, between January and March 2021. The study used a sample of full-time university students from all academic years.” (See line 78-80, page 2)

- What faculties are these students from? medical, nursing, dietetics, others?

We thank the Reviewer for this comment. The questionnaire that we used in the current study did not include question about faculty, but, as we said in the previous comment, we sent the questionnaire through university e-mail to all university students from all faculties. Thus, we assumed that the participants are from different faculties.

- Was the questionnaire validated, any test-retest?

We thank the Reviewer for this comment and the chance to clarify. To address this comment, we have added a sentence in the questionnaire design section and in the references, as follows:

The questions included in the survey were developed using data from previous studies that examined the intake of snacks and lifestyle behaviors of university students in different countries [30-32].” (See lines 92-94, page 2).

  1. The summary is imprecise (repeating the aim of the study and the results are not the conclusions) and requires improvement.

We thank the reviewer for the valuable comments. The conclusion has been rewritten (see page 10 and 11). In the current study we investigated lifestyle behaviours and type of snacks related to risk of obesity among university students. Snacks consumed contribute to the intake of daily energy and nutrients and thus affect the overall quality of diet. Most highly consumed snacks among the study participants were considered unhealthy item expect for fruits, vegetables, and nuts. Consumption of potato chips, popcorn, and biscuits as snack items was highly associated with obesity risk, while salad intake was associated with decreased obesity risk. The university environment was associated with reduced consumption of healthy type of snacks such as fruit and vegetables, nuts, and dairy products. Health education plans and programs among university students are crucial to encourage healthy eating habits, including eating breakfast and frequent consumption of healthy snacks. Universities should also focus on the food environment at campuses and ensure accessibility to healthy snacks at reasonable prices, which is essential in developing healthy dietary behaviours and decreasing the high rates of obesity in young adults, and ultimately improving public health.”

Round 2

Reviewer 1 Report

I have no comments. All suggestions and comments have been included in the manuscript. The authors provided exhaustive answers.